# The Essential Role of PCR and PCR Panel Size in Comparison with Urine Culture in Identification of Polymicrobial and Fastidious Organisms in Patients with Complicated Urinary Tract Infections

**DOI:** 10.3390/ijms241814269

**Published:** 2023-09-19

**Authors:** Xingpei Hao, Marcus Cognetti, Chiraag Patel, Nathalie Jean-Charles, Arun Tumati, Rhonda Burch-Smith, Mara Holton, Deepak A. Kapoor

**Affiliations:** 1P4 Diagnostix, Pine Brook, NJ 07058, USA; marcus.cognetti@p4dx.com (M.C.); chiraag.patel@p4clinical.com (C.P.); nathalie.charles@p4clinical.com (N.J.-C.); arun.tumati@p4clinical.com (A.T.); rhonda.burchsmith@p4dx.com (R.B.-S.); 2Anne Arundel Urology, Annapolis, MD 21401, USA; mholton@aaurology.com; 3Solaris Health Holdings, LLC, Fort Lauderdale, FL 33394, USA; 4The Icahn School of Medicine at Mount Sinai, New York, NY 10029, USA

**Keywords:** complicated urinary tract infection, fastidious organism, molecular testing, PCR, polymicrobial organism, urine culture

## Abstract

Complicated urinary tract infections (cUTIs) are difficult to treat, consume substantial resources, and cause increased patient morbidity. Data suggest that cUTI may be caused by polymicrobial and fastidious organisms (PMOs and FOs, respectively); as such, urine culture (UC) may be an unreliable diagnostic tool for detecting cUTIs. We sought to determine the utility of PCR testing for patients presumed to have a cUTI and determine the impact of PCR panel size on organism detection. We reviewed 36,586 specimens from patients with presumptive cUTIs who received both UC and PCR testing. Overall positivity rate for PCR and UC was 52.3% and 33.9%, respectively (*p* < 0.01). PCR detected more PMO and FO than UC (PMO: 46.2% vs. 3.6%; FO: 31.3% vs. 0.7%, respectively, both *p* < 0.01). Line-item concordance showed that PCR detected 90.2% of organisms identified by UC whereas UC discovered 31.9% of organisms detected by PCR (*p* < 0.01). Organism detection increased with expansion in PCR panel size from 5–25 organisms (*p* < 0.01). Our data show that overall positivity rate and the detection of individual organisms, PMO and FO are significantly with PCR testing and that these advantages are ideally realized with a PCR panel size of 25 or greater.

## 1. Introduction

Urinary tract infection (UTI) refers to an infection of any part of the urinary tract, including the urethra, bladder, ureter, and kidneys. UTIs are extremely common, affecting more than 10 million patients in the United States each year [1]. Most UTI are uncomplicated type, defined as a lower tract infection without associated anatomic abnormalities and resolved with appropriate treatment [2,3,4]. Complicated urinary tract infections (cUTIs) are generally accepted to be any UTI that fails to meet the criteria for simple UTI, and as such, encompasses a wide variety of conditions, including patients who have recurrent uncomplicated UTI despite adequate treatment but have prolonged symptoms or rapid recurrence of symptoms [5], those that present with underlying anatomic abnormalities, concomitant comorbidities (urologic or systemic), as well as recent hospitalization or genitourinary instrumentation [6,7]. cUTIs are associated with increased morbidity and frustration for patients, and it is often quite challenging for clinicians to develop an effective plan for eradication and to prevent rapid recurrence, especially when the cUTI is caused by polymicrobial organisms (PMOs) and/or fastidious organisms (FOs) [8,9,10,11,12]. It is estimated that over 3.3 million patients develop cUTIs annually [9]. In 2018, reports suggest that there were over 600,000 hospitalizations due to cUTIs, representing an increase of more than six-fold over two decades in the United State [13,14], costing healthcare billions’ dollars annually [14].

Empiric treatment of UTIs is frequently initiated prior to culture results without identification of the pathogen(s) and susceptibility profile [2,15]. While this is usually adequate and appropriate for a simple UTI, the same cannot be said for cUTIs. In these patients, the symptoms often (1) persist or recur rapidly, requiring short interval re-evaluation and re-treatment, or (2) progress in severity, resulting in more serious or longer-term complications, including mortality [9], or (3) have an adverse impact on the GU tract or other systems. Furthermore, patients with incomplete resolution or rapid recurrence of symptoms after antibiotic treatment are at increased opportunity for antibiotic resistance. Finally, overutilization of broader spectrum antibiotics in these patients, in an attempt to avoid the possibility of bacterial resistance in the acute episode, contributes to antibiotic side effects and the continued development of antibiotic resistance of organisms [16]. Rapid and accurate identification of the pathogens involved in cUTIs is essential to mitigate these dependent considerations.

Multiple modalities have been developed to aid in the diagnosis of UTI, including urine dipstick, urine microscopy, and urine culture (UC) [17]. UC has been regarded as the gold standard to identify the pathogen(s) and to generate an antimicrobial susceptibility profile for proper treatment. UC is a lengthy process (typically between 24 to 48 h) requiring both selective and nonselective media, as well as experienced technical personnel for interpretation of culture results [18,19]. Some slow-growing pathogens require longer incubation time to grow [19], while some intracellular bacteria and longer-term intracellular reservoirs may fail to grow in UC entirely [20]. It has been reported that 25 to 30% of urine cultures produce negative results in symptomatic patients [21,22]. Finally, in certain polymicrobial infections, where two or more competitive bacteria are present, one organism may outgrow the other in culture, thus leading to identification and reporting of only a single organism. This missed diagnosis leading to failure to treat the ‘secondary’ bacterial pathogen may be associated with incomplete resolution and rapid recurrence of symptoms and necessitate prolonged or multiple treatment courses. These scenarios point to the necessity of developing more accurate and rapid modalities to identify pathogens involved in UTIs.

Several pioneer studies have suggested that recently developed syndromic polymerase chain reaction (PCR) molecular assays have decreased turn-around time (TAT) and are more effective in identifying PMOs and FOs when compared to conventional UC [23,24,25,26,27]. These studies were limited as the patient population was not confined to those clinically identified with cUTIs. The current study presents a retrospective review of 40,029 patients whose urine specimens were tested concomitantly by both UC and PCR with the aim to determine whether the addition of urinary PCR testing to conventional UC is of clinical utility specifically in this difficult-to-treat subset of the patients. In addition, we seek to determine whether and to what degree PCR panel size increases sensitivity in the detection of potential urinary tract pathogens in patients with a cUTI, both overall and when PMOs and FOs are present.

## 2. Results

### 2.1. Patients’ Age and Gender Distribution

The mean patient age among 36,325 patients with clinically diagnosed cUTIs with complete demographic information was 64.4 (range 2–107) years. Age differential was influenced by gender, with 21,589/36,325 (59.4%, mean 63.7 years, range 2–107 years) female while 14,736/36,325 (40.6%, mean 65.7 years, range 2–103 years) were male (difference 18.4%, *p* < 0.01). Most patients were older than 60, with 24,320/36,325 (67.0%) vs. 12,005/36,325 (33.0%) (older vs. younger than age 60, difference 34.0%, *p* < 0.01). Although significant for both age groups, the female to male ratio was greater for patients under age 60. The number of females vs. males was 7735/12,005 (64.4%) vs. 4270/12,005 (35.6%) of patients <60 compared to 13,854/24,320 (57.0%) female and 10,466/24,320 (43.0%) male for those patients 60 and older.

### 2.2. Overall UC and PCR Positivity Rates

The overall results stratified by test type and result for 36,586 patient specimens included in analysis are illustrated in Table 1. The overall positivity rate was 33.9% and 52.3% for UC and PCR, respectively (difference 18.4%, *p* < 0.01). In addition, 20.4% of UC negative (-) specimens were PCR positive (+) while 1.9% of PCR- specimens were UC+. Among 711 PCR-/UC+ specimens, 50.9% (362/711) were due to organisms not included on the PCR panel.

### 2.3. Analysis of Polymicrobial and Fastidious Organisms Detected by UC and PCR

The distribution of organisms per specimen is illustrated in Table 2. Of 12,393 UC+ specimens, 11,945 (96.4%), 445 (3.6%), and 3 (0.0%) grew 1, 2, and 3 organisms, respectively. In contrast, of 19,129 PCR+ specimens, 10,299 (53.8%), 4751 (24.8%), 2244 (11.7%), and 1835 (9.6%) identified 1, 2, 3, and 4 or more organisms, respectively. The difference in number of organisms found for 1 (42.5%), 2 (21.2%), and 3 (11.7%) organisms per specimen was significant (*p* < 0.01) for each category. In total, there were 899/12,844 (7.0%) organisms found in 448/12,393 (3.6%) of UC+ specimens which identified cUTIs with PMOs; simultaneously, 24,976/35,275 (70.8%) of organisms identified in 8830/19,129 (46.2%) of PCR+ specimens were found to have PMOs. Both the difference in PMO detection in specimens (42.6%) and number of organisms found (63.8%) between UC and PCR was significant (*p* < 0.01).

We found that 0.7% (92/12,393) of patient specimens representing 0.7% (92/12,844) of total organisms identified on UC+ met our laboratory’s criteria as FO. There was a total of four FOs isolated on UC, two species (*A. urinae* and *S. oralis*, 66/92 and 9/92 of FOs grown on UC, respectively) were on the PCR panel while two Enterococcus species (*E. galliflavus* and *E. gallinarum*, 3/92 and 14/92 of FOs isolated on UC, respectively) were not. Conversely, a total of 5994/19,129 (31.3%) of PCR+ specimens had FOs comprising 7473/35,275 (21.2%) of all organisms found on PCR. Both the difference in specimen and organism detection (30.6% and 20.5%, respectively) between UC and PCR were significant (*p* < 0.01). Although perhaps not traditionally identified as FOs (e.g., viruses), several organisms identified on PCR did not grow at all in UC. In total, 19/45 (42.2%) of organisms identified on PCR were not found on any UC specimens. These accounted for 7715/35,275 (21.9%) of all organisms identified by PCR.

### 2.4. Organism by Organism Line-Item Concordance Comparison of 45 Organisms Identified by PCR vs. UC

Table 3 lists depicts the organism by organism line-item concordance between PCR and UC [28] for the 45 organisms present on the PCR panel (from high to low frequency of PCR detected organisms). Overall, 90.2% of organisms that grew in UC were detected by PCR whereas UC revealed 31.9% of organisms detected by PCR (difference 58.3%, *p* < 0.01). PCR significantly outperformed UC in organism detection for all but three pathogens: *A. baumannii, C. freundii*, and *P. agglomerans*; PCR did detect more of the former than UC, but the difference was not significant, while the latter two were the only two organisms on the PCR panel where UC outperformed PCR, although neither difference was significant. In total, there were 362 specimens with a total of 362 organisms representing 22 genera and 53 species that were identified on UC but not on the PCR panel; these represented 2.9% of specimens and 2.8% of organisms found on UC. Four organisms isolated on UC were not subclassified into species level (three Gram-negative rods and one Gram-positive coccus), it could not be determined if they were among the 45 organisms on the PCR panel. As summarized in Table 4, of organisms isolated on UC but not found on the PCR panel, the top 10 genera accounted for 75.5% (40/53) of identified species and 93.1% (337/362) of this subset of specimens.

### 2.5. Impact of the PCR Panel Size on Organism Detection

Table 5 demonstrates the extent to which the PCR panel size affected the detection of organisms on PCR relative to the detection rate on UC. As this is sorted by frequency of detection by PCR, the 362 UC detected organisms not on the PCR panel are added to each column, as they would not be detected regardless of panel size (the 362 organisms include 4 organisms isolated but without species identified on UC). In total, the 45-organism panel detected 11,263/12,844 (87.3%) organisms found on UC. Sorted by frequency of organism detection by PCR, with increased panel size, the PCR non-detection rate decreased significantly (*p* < 0.01) from 34.4% (5-organism panel) to 13.6% (25-organism panel); the difference in detection rate between each hypothetical panel was significant (*p* < 0.01). The total reduction with PCR panel size expansion from 5 to 25 organisms was significant as well (*p* < 0.01). Although there was a nominal decrease in PCR non-detection rates with panel sizes of greater than 25, these differences were not significant; despite this, the small absolute decrease (0.9%) in PCR non-detection rate from 25 to 45 organisms was significant (*p* = 0.03)

Importantly, assessing the impact of PCR panel size on organism detection based on UC artificially excludes organisms not detected by UC (in our case, 19/45 organisms on the panel) and given the relative paucity of such specimens on UC, does not allow for stratification based on detection of PMOs and FOs. Table 6 illustrates the impact of PCR panel size on organism detection within the PCR panel, sorted by number of organisms detected by PCR, and further stratified by UC result. Overall, the PCR detection rates of all organisms, PMOs increased significantly from 5- to 25-organisms while the PCR detection rates of FO increased significantly up to a 20-organism panel (Table 6, all *p* < 0.01). To achieve detection rates of 95% or greater required a panel size of 25 organisms; from 5-organism panel to 25-organism panel, the PCR detection rates increased from 79.8% to 98.2% for all pathogens, from 54.1% to 97.7% for PMOs and from 61.7 to 97.6% for FOs (all *p* < 0.01). There was no significant difference between panels with 25 organisms or greater for all organisms and PMOs, and for over 20 organisms for FOs.

When stratified by UC results, other than for detection of FO on UC+ specimens and overall organism detection in UC- specimens, a panel size of 25 organisms was required to detect 95% of organisms; for the other two classifications, a 20-organsim panel met this requirement. When stratified by culture result, incremental increases of 5 organisms from 5 to 25 in UC+ was significant for all organisms and PMO (*p* < 0.01) and from 5 to 20 organisms for FO (*p* = 0.05). For UC- specimens, an incremental increase from 5 to 20 organisms was significant for all specimens and FOs (*p* < 0.01 and =0.05, respectively) while for this subset of specimens, significant incremental increase in organism detection was noted up to 25 organisms for PMOs (*p* = 0.03).

## 3. Discussion

This study represents the largest analysis of the use of PCR in cUTIs to date, comprising 36,586 specimens received over a period of 26 months from independent urologic offices located in 19 different states. This large study provides comprehensive data to profile microbial infections in cUTIs patients, and to evaluate whether PCR testing provided clinically relevant additional data that could potentially assist in management of these difficult-to-treat patients.

As with prior reports, we found that the preponderance of patients with cUTIs were female; and this large dataset also showed that patients older than 60 represented two times the patients tested than those <60 (67% vs. 33%) [8]. The preponderance of female patients was markedly higher in the younger age group; in fact, almost 2/3 of those patients less than 60 (64.4%) were female. We postulate that this may be due to progression to cUTI through relapse/recurrence of simple UTI in younger women, of which the higher incidence compared to male counterparts has been well established [5].

Our data showed that PCR was more sensitive in the identification of pathogens in patients with presumed cUTI than UC at both the specimen and organism level. Our findings quantify, for this subset of patients, prior reports suggesting that UC seldom grows multiple organisms, likely due to some combination of media requirement, organism growth suppression, or competition. We found that the detection rate of PMOs was significantly increased from 3.6% in UC+ to 46.2% in PCR+ specimens. The FO detection rate was also significantly increased from 0.7% in UC+ to 31.3% in PCR+ specimens. Given the challenges in treating these patients, the inability of conventional UC to detect potential pathogens points to the need for alternative testing methodologies; our data suggest that PCR testing may play an important role in this niche.

Although comparison of overall organism detection and test positivity is helpful, this is not meaningful to assess test utility on an individual organism or specimen level; to do this, it is necessary to compare the detection of each organism on a line-item basis for each specimen. We used the method of organism-by-organism line-item concordance elucidated by Kapoor et al. [28] to compare all 45 organisms in the PCR panel with corresponding organisms identified in UC. This showed that overall, PCR was 274.2% more effective than UC, detecting 90.2% of potential pathogens detected by UC whereas UC identified 31.9% of potential pathogens discovered by PCR. Concordance analysis found that *E. coli* and *E. faecalis* were the top two most frequently detected organisms by both UC and PCR, which is consistent with prior reports [29,30,31], accounting for 57.9% and 34.6% of organisms detected, respectively. For *E. coli* and *E. faecalis*, PCR was 134.0% and 242.9% more sensitive than UC. Further concordance analysis of the subsequent 8 organisms revealed that 3 of the top 10 organisms (30%) most commonly detected by PCR (*S. pasteurianus, U. urealyticum,* and *A. schaalii*) were not isolated on UC at all. The necessity to detect FOs in cUTI patient was exemplified by the fact that *A. urinae, U. urealyticum,* and *A. schaalii* were the third, seventh, and eighth most frequently detected organisms on PCR, while *S. pasteurianus* (which did not grow in UC at all) was the fourth most common organism detected by PCR. FOs, including *A. urinae*, *A. schaalii, M. genitalium*, and *U. urealyticum,* as well as *S. pasteurianus*, have been increasingly recognized as causative or contributory agents in cUTIs [6,32,33,34,35,36,37,38].

Of the 25/45 organisms on the PCR panel in which both tests had at least one specimen in common, UC failed to grow 54.5% (12,473/27,418) of organisms detected on PCR. Interestingly, although there were 9 and 142 instances where UC and PCR were positive for *S. oralis*, respectively, there were no specimens where both tests were positive. Conversely, PCR was not without limitations: there was at least one instance for every organism detected by UC in this study that PCR performed on the same specimen failed to detect the organism. Indeed, there were two organisms where UC was superior to PCR in organism detection, albeit not significantly. This suggests that both UC and PCR are complimentary tools to detect potentially pathogenic organisms in patients with cUTI.

We also sought to determine what the optimal PCR panel size would be to ensure organism detection in clinical applications. Given that this is a difficult-to-treat patient population, we sought to determine if there was a threshold in which positivity rates would exceed 95% in total, and for the detection of PMOs and FOs individually. We further stratified this by whether the specimen was UC+ or UC-. Our data showed that the PCR detection rates of all organisms, PMOs, and FOs were significantly increased from 5- to 25-organism panels. The 25-organism PCR panel detection rates of all organisms, PMOs, and FOs reached 98.2%, 97.7%, and 97.6%, respectively. Even though the small improvement in specimen detection was significant with the expansion of panel size, there were no significant incremental differences in specimen positivity or the detection of PMOs or FOs. Importantly, these results were replicated when stratified by culture results; either a 20- or 25-organism panel was necessary to achieve adequate detection of potential pathogens.

Despite compelling evidence of PCR sensitivity and specificity, as with all laboratory tests, it is critical that the data are considered and interpreted in the context of a patient’s presentation and the provider’s clinical experience and judgement. For this reason, PCR, as with any other test, should be considered information that is accretive to the clinician’s decision-making process. As such, the treating physician may opt to initiate, alter, continue, or discontinue treatment based on the PCR findings. In our series, over a fifth of patients had potential pathogens identified on PCR despite negative urine cultures, crucially important data in this difficult-to-diagnose and -treat subset of patients. It is for this reason that even a negative PCR is of substantial clinical utility—as a negative UC is far from a guarantee that no pathogens are present, a confirmatory negative PCR can suggest to the clinician that antimicrobial therapy may be discontinued or, more importantly, there may be an alternative pathologic process as an etiologic factor which may require additional evaluation.

An additional potential limitation of this study is the identification of patients with cUTIs. We rely on the clinical practice that refers to us to develop internal guidelines for the use of diagnostic tests; as a national reference laboratory, we do not practice medicine nor make these clinical decisions. As such, it is possible that some of the patients did not meet the internal practice guidelines for cUTIs. To address this limitation, we restricted our analysis not only to those practices whose guidelines included common parameters used to identify patients with cUTI, but who confirmed they have internal quality management processes in place to monitor physician compliance with these guidelines.

The enhanced detection of organisms by PCR also may pose unique challenges. For example, both UC and PCR demonstrated *S. epidermidis* in both women and men, but PCR detection was 236.4% that of UC. *S. epidermidis* is considered a commensal organism part of the normal flora of the urogenital tract and skin and is generally not considered a urinary pathogen. However, *S. epidermidis* is also an opportunistic pathogen in immunocompromised patients, such as those with kidney transplants or with indwelling catheters [39]. *S. epidermidis* was reported as a causative organism of UTIs in children with underlying urinary tract abnormalities including bladder diverticula and vesicoureteral reflux [40]. Furthermore, although infections due to PMOs have been identified as causative in patients with a cUTI, it may be difficult when multiple organisms are detected on either UC or PCR to either precisely identify which is the pathogenic organism or what may constitute a contaminant. In addition, PCR may potentially detect non-viable organisms in a patient’s urine given its enhanced sensitivity. Thus, the PCR results should be cautiously interpreted in conjunction with the patient’s symptoms and any known anatomic or immunologic abnormalities by a provider who can appropriately correlate the laboratory information with clinical findings to optimize the treatment of the patient.

Both UC and PCR have limitations in detecting pathogens. The administration of antibiotics may inhibit the growth of microorganisms in UC, leading to false negative results. A major contributing factor in our series for the differential in organism detection between UC and PCR is fastidious pathogens including *A. urinae*, *Mycoplasma* spp., and *Ureaplasma* spp., as well as intracellular pathogens which may not grow in UC [41]. However, variations in microbiological technique may result in different growth patterns in different laboratories; while we identified 13 organisms that we would consider fastidious (11 that were on the PCR panel and 2 that were not), a different laboratory may have different organisms that they find difficult to identify on routine UC. These limitations can, at least partially, be overcome with the utilization of PCR, as short-term antibiotic use and growth requirements would not be expected to impact PCR’s ability to detect these organisms.

While PCR was extremely sensitive overall in organism detection and markedly more so than conventional UC (52.3% vs. 33.9%), PCR has intrinsic limitations which should be highlighted. PCR testing sensitivity is fundamentally restricted to the pre-defined panel of organism specific primers contained in the PCR microarray. We offset this concern by using a multi-organism panel; indeed, part of the focus of this study was to determine optimal panel size for pathogen detection. Other factors which impact PCR sensitivity, including technical limitations, must be considered. These include factors including reagent quality, PCR inhibitors in the samples, low extraction efficiency, or bubble formation in the open array chip. These limitations were mitigated by the utilization of a single, large-volume commercial lab with extensive experience in PCR testing that incorporates extensive technical training and quality control measures to limit these laboratory pitfalls. Even with this, we observed a marked variance in detection of *C. freundii,* notable as one of only two organisms where UC+ exceeded PCR+ on a line-item concordance analysis (the other being *P. agglomerans*). Fully characterizing the significance and reproducibility of this deviation merits further investigation and could be related to any number of factors including unique laboratory and/or reagent characteristics. However, potential consideration in this scenario also includes PCR’s total dependence on defined bacterial genetic target sequences. In the compressed evolutionary timeline of bacteria, it is possible that these sequences may be prone to more rapid change and degradation of detection over time. Overall, there were 711 incidences in which an organism was identified on culture which was not detected by PCR. While nearly 50% of the time this was due to the organism’s non-inclusion on our PCR pathogen detection list, in most cases, it was individual species of a PCR-detected genus that were not identified, due to the high specificity of speciated PCR primers.

These findings highlight key considerations in the use of PCR testing in cUTIs, those of panel size and organism selection. We continuously evaluate data from our client practices and review literature to ensure that our panel provides optimal actionable clinical data to physicians that use our services. For example, based on literature published early this year [42], we no longer include *S. oralis* on this panel and have replaced it with human betaherpesvirus 7 (HHV7) [43]. That said, it may be that there are regional variations in pathogen distribution which may impact panel selection. This is particularly important should a lab opt for a smaller PCR panel—our data suggest that a 25-organism panel should be adequate for potential pathogen detection; however, selection of the organisms for that panel is crucial to facilitate accurate clinical decision making. Further investigation into the impact of PCR on urology resource utilization and patient outcomes is ongoing.

## 4. Materials and Methods

### 4.1. Patient Selection

In total, 40,029 patients with midstream clean-catch or indwelling urinary catheter collected urine samples from multiple community based urological offices located in 19 states tested by both UC and PCR in a national reference laboratory, P4 Diagnostix (Pine Brook, NJ, USA), from 1 January 2021, to 28 February 2023, were retrospectively reviewed. Analysis was restricted to practices that have established internal clinical criteria for PCR use; although precise criteria varied across practices, certain clinical criteria were common to all groups and are listed in Table 7. Based on this, 3443 patients were excluded from analysis as we were unable to verify the existence of these protocols at the practice level.

Of 36,586 patients who met clinical criteria and were included in the result analysis, 261 (0.7%) were excluded from demographic analysis either because age or gender information was not available. In total, 36,325 patients were available for demographic analysis. The study was conducted in accordance with the Declaration of Helsinki and was IRB exempted from ethical review under 45CFR 46.101(b)4 [44].

### 4.2. Urine Culture

Each patient’s urine was collected in a BD Urine C&S Transport 4.0 mL Vacutainer Tube through either midstream clean catch or indwelling urinary catheter. All urine specimens were shipped to P4 Diagnostix in Pine Brook, New Jersey via an overnight service. Once received, the urine specimens were first accessioned and then processed for UC and subsequently PCR. Then, 1 µL urine was streaked with a standard disposable loop on each side of a biplate with TSA with 5% sheep blood and MacConkey agar plate (ThermoFisher Scientific, Carlsbad, CA, USA). The plates were incubated at 35–37 °C for at least 18 h, then visually inspected for colony growth and assessed for quantity and morphology. Cultures with no visible growth were further incubated for an additional 24 h and re-inspected. Isolated colonies were further incubated on Vitek 2 (BioMerieux, Durham, NC, USA) overnight to identify microbial species level and profile antimicrobial susceptibility according to manufacturer’s instructions.

### 4.3. Molecular PCR Testing

#### 4.3.1. Urine DNA Extraction

First, 800 µL of each patient’s urine was aliquoted into each well of a 96-deep well plates, sealed with foil, and then centrifuged. Then, 700 µL supernatant was removed from each well and 50 µL of enzyme mix for lysis was added to each well with concentrated urine and an extraction negative control (100 µL nuclease-free water) and incubated at 65 °C for 20 min on KingFisher Flex (ThermoFisher Scientific, Carlsbad, CA, USA). Subsequently, 240 µL of binding solution containing DNA bead, proteinase K, and XENO (Internal control, ThermoFisher) was added into each well and then processed on KingFisher Flex for protein digestion and DNA extraction for 30 min.

#### 4.3.2. PCR Analysis

First, 2.5 µL of extracted urine DNA and 2.5 µL of master mix (TaqMan^TM^ OpenArray^TM^ Real Time PCR Master Mix, Applied BioSystems, ThermoFisher, Carlsbad, CA, USA) were separately added into a 384-well plate and mixed well, and then spotted onto OpenArray^TM^ Chip in duplicate for each sample and loaded onto the Applied Biosystems^®^ QuantStudio™ 12K Flex Real-Time PCR System (Applied BioSystems, ThermoFisher) for DNA amplification as instructed by the manufacturer. A positive control (Taqman^TM^ Comprehensive Microbiota Control 2, Applied BioSystems, ThermoFisher) and a negative control (100 µL nuclease free water) were included in each chip. The QuantStudio 12 Flex Software Program (Applied BioSystems, ThermoFisher) was used for PCR analysis and the data were further analyzed for quality assurance. Positive PCR (PCR+) was defined as any organism that displayed a cycle threshold of less than or equal to 29.75 (as determined by our Limit Of Detection corresponding to a pathogen detection level of 1 × 10^3^ copies of pathogen per mL of sample with 2 Sigma analysis), in duplicate (StDev was less than or equal to 2.0), with an amplification score of 1.24 and a minimum Cq confidence score of 0.8 (recommended by ThermoFisher), and the simultaneous presence of Xeno as a determinant of PCR inhibition.

### 4.4. Targeted Organisms

This custom assay evaluated a total of 45 organisms (Table 8). The sequences of the primers and probes are not available due to being proprietary to Fisher but the assay IDs corresponding to each organism are provided in Table 8. In our laboratory, a PMO is defined as two or more organisms in one patient’s urine specimen identified by either PCR or UC. FO refers to any organism that, in our laboratory, requires specific nutrients and atmospheric conditions including temperature, oxygen, and carbon dioxide to grow on agar plates (highlighted in blue in Table 8).

### 4.5. Data Analysis

The overall positive and negative detection rates were calculated based on the ratio of the sum of the positive and negative cases to the total number of cases analyzed. The total number of specimens containing PMOs and FOs were tallied for each test compared to the total number of positive cases.

Organism line-item concordance was conducted to compare the efficiency of PCR with UC to detect individual organism [28]. This was performed by identifying each organism found in every specimen and assessing if the same organism was found on both tests. The ratio of organisms common to both tests to total organisms found by each test was compared and tested for significance, as were total organisms detected. For example, a concordance ratio of 100% indicates that all organisms found on one test would be detected by the other, while a ratio of 0% indicates that the test in question did not identify any of that organism found on the corresponding test. As, by definition, organisms not on the PCR panel cannot be identified, the analysis was restricted to the 45 organisms on the PCR panel.

To analyze the impact of PCR panel size on PCR detection rate, we analyzed both the PCR detection of organisms detected by UC as well as compared the organism detection rate within the PCR panel. To do this, the 45 pathogens detected by PCR were divided into 9 hypothetical panels based on frequency of PCR organism detection with a 5-organism incremental increase in each panel and sorted by the frequency of organism detection via PCR. In the case of UC, the PCR detection rates in each panel were analyzed for all organisms by comparing the organism positivity for each hypothetical panel against the total number of organisms found on UC, while for PCR, the organism positivity for each incremental panel was compared to the organism positivity for the full 45-organism panel. In addition to overall organism detection, we analyzed the relative detection of PMOs and FOs for each hypothetical panel and further stratified the analysis via urine culture result; this was not done in the comparison of UC and PCR due to small sample size.

Statistical analysis of overall specimen positivity and detection of PMO and FO was performed using the chi-square test. Comparison of total line-item concordance was determined using Student’s paired *t*-test. The difference in individual organism line-item concordance, the difference in number of PMOs detected on each test, and the effect of incremental increase in panel size on organism detection were determined with a two-proportion z-test. Statistical analysis was performed using GraphPad Prism version 9.1.5 (GraphPad Software, San Diego, CA, USA) and Microsoft Excel (Microsoft^®^ Excel^®^ for Microsoft 365 MSO (Version 2307 Build 16.0.16626.20170) 64-bit).

## 5. Conclusions

The subset of patients with a UTI that present with or develop a cUTI represents a challenging clinical management problem, one which can result in substantial patient morbidity and consumption of significant healthcare resources. While there are ample data showing that PCR outperforms UC in organism detection in UTIs, our study is the first to specifically analyze this differential in patients with cUTIs, and to quantify this differential for patients with PMOs and FOs. We also found that in addition to overall positivity rates, individual organism line-item concordance strongly favored PCR over UC for virtually all organisms on the PCR panel, with nearly half of the organisms on the PCR panel not isolated on any UC. Despite the superiority of PCR over UC, even a 45-organism PCR panel did not capture 100% of organisms isolated on UC, nor was there 100% concordance for any organism found on both tests. We conclude that for patients with a cUTI, UC and PCR are complementary tests that should be used in conjunction to provide clinicians with complete data on which to make diagnostic and management decisions for this difficult-to-treat subset of patients.

## Figures and Tables

**Table 1 ijms-24-14269-t001:** UC and PCR results by specimen.

PCR	Urine Culture		
	Negative	Positive	Total
Negative	16,746 (45.7%)	711 (1.9%)	17,457 (47.7%)
Positive	7447 (20.4%)	11,682 (31.9%)	19,129 (52.3%)
Total	24,193 (66.1%)	12,393 (33.9%)	36,586 (100.0%)

**Table 2 ijms-24-14269-t002:** Organisms per specimen found by UC and PCR.

Test	Number of Organisms/Specimen (n and % Total)					Total Organism Count
	1	2	3	≥4	Specimen Count	
UC	11,945 (96.4%)	445 (3.6%)	3 (0.0%)	0 (0.0%)	12,393 (100.0%)	12,844
PCR	10,299 (53.8%)	4751 (24.8%)	2244 (11.7%)	1835 (9.6%)	19,129 (100.0%)	35,275
*p*	<0.01	<0.01	<0.01	<0.01		

**Table 3 ijms-24-14269-t003:** Line-item concordance analysis for organisms on PCR panel.

PCR Panel	Organism	UC+	PCR+	Both+	UC Ratio	PCR Ratio	*p*
1	*E. coli*	5384	7219	5122	95.1%	71.0%	<0.01
2	*E. faecalis*	2056	4989	1722	83.8%	34.5%	<0.01
3	* A. urinae *	66	3698	54	81.8%	1.5%	<0.01
4	*S. pasteurianus*	0	3622	0	n/a	0.0%	n/a
**5**	*K. pneumoniae*	1727	2245	1646	95.3%	73.3%	<0.01
6	*S. haemolyticus*	68	1909	60	88.2%	3.1%	<0.01
7	* U. urealyticum *	0	1359	0	n/a	0.0%	n/a
8	* A. schaalii *	0	1108	0	n/a	0.0%	n/a
9	*S. Lugdunensis*	27	1076	22	81.5%	2.0%	<0.01
**10**	*S. epidermidis*	423	1000	356	84.2%	35.6%	<0.01
11	*S. agalactiae*	302	918	255	84.4%	27.8%	<0.01
12	*P. aeruginosa*	399	794	377	94.5%	47.5%	<0.01
13	*P. mirabilis*	513	704	426	83.0%	60.5%	<0.01
14	* A. omnicolens *	0	524	0	n/a	0.0%	n/a
**15**	*E. cloacae*	296	442	271	91.6%	61.3%	<0.01
16	*E. faecium*	119	379	94	79.0%	24.8%	<0.01
17	*C. glabrata*	0	369	0	n/a	0.0%	n/a
18	* C. riegelii *	0	354	0	n/a	0.0%	n/a
19	*M. morganii*	113	348	104	92.0%	29.9%	<0.01
**20**	*C. albicans*	27	346	21	77.8%	6.1%	<0.01
21	*K. oxytoca*	282	345	236	83.7%	68.4%	<0.01
22	*S. aureus*	177	290	154	87.0%	53.1%	<0.01
23	* M. hominis *	0	250	0	n/a	0.0%	n/a
24	*E. aerogenes*	164	222	140	85.4%	63.1%	<0.01
**25**	*C. koseri*	92	145	85	92.4%	58.6%	<0.01
26	* S. oralis *	9	142	0	0.0%	0.0%	n/a
27	*S. marcescens*	52	95	48	92.3%	50.5%	<0.01
28	*C. amalonaticus*	13	75	12	92.3%	16.0%	<0.01
29	*C. freundii*	120	74	22	18.3%	29.7%	0.07
**30**	*C. parapsilosis*	0	56	0	n/a	0.0%	n/a
31	*S. saprophyticus*	18	38	15	83.3%	39.5%	<0.01
32	*A. baumannii*	19	33	11	57.9%	33.3%	0.08
33	*P. stuartii*	10	30	9	90.0%	30.0%	<0.01
34	* M. genitalium *	0	20	0	n/a	0.0%	n/a
**35**	*HHV6*	0	10	0	n/a	0.0%	n/a
36	* C. trachomatis *	0	8	0	n/a	0.0%	n/a
37	*HSV2*	0	8	0	n/a	0.0%	n/a
38	*CMV*	0	6	0	n/a	0.0%	n/a
39	* C. urealyticum *	0	6	0	n/a	0.0%	n/a
**40**	*HSV1*	0	5	0	n/a	0.0%	n/a
41	*P. agglomerans*	6	4	1	16.7%	25.0%	0.75
42	* N. gonorrhoeae *	0	4	0	n/a	0.0%	n/a
43	*T. vaginalis*	0	3	0	n/a	0.0%	n/a
44	*S. pyogenes*	0	2	0	n/a	0.0%	n/a
**45**	*M. tuberculosis*	0	1	0	n/a	0.0%	n/a
	Total	12,482	35,275	11,263	90.2%	31.9%	<0.01

Note: PCR panel lists the frequency of organisms detected by PCR; Panels 5, 10, 15, 20, 25, 30, 35, 40, 45 were bolded. The fastidious organisms are highlighted in blue. UC ratio is calculated from the number of Both UC+ and PCR+ specimens divided by the total of UC+ specimens while PCR ratio from the number of Both+ specimens divided by the total of PCR+ specimens.

**Table 4 ijms-24-14269-t004:** Specimen and species count of top 10 genera found on UC but not PCR.

Genus	Species (N)	Species (% Total)	Specimens (N)	Specimens (% Total)
Staphylococcus	11	20.8%	113	31.2%
Providencia	1	1.9%	38	10.5%
Raoultella	2	3.8%	37	10.2%
Pseudomonas	4	7.5%	30	8.3%
Proteus	3	5.7%	29	8.0%
Citrobacter	4	7.5%	26	7.2%
Enterococcus	4	7.5%	24	6.6%
Serratia	5	9.4%	20	5.5%
Streptococcus	4	7.5%	13	3.6%
Acinetobacter	2	3.8%	7	1.9%
Total	40	75.5%	337	93.1%

**Table 5 ijms-24-14269-t005:** Impact of the PCR panel size on non-detection of organisms found on UC.

PCR Panel Size	5	10	15	20	25	30	35	40	45
UC+ Detected on Panel	8544	8982	10,311	10,530	11,145	11,227	11,262	11,262	11,263
UC+ Organisms Not Detected	4300	3862	2533	2314	1699	1617	1582	1582	1581
% UC+ Not Detected	34.4%	30.9%	20.3%	18.5%	13.6%	13.0%	12.7%	12.7%	12.7%
*p* value (to prior panel)		<0.01	<0.01	<0.01	<0.01	0.13	0.51	1.00	0.98

**Table 6 ijms-24-14269-t006:** Impact of PCR panel size on the PCR identification of all organisms, PMOs, and FOs.

Culture Result	Specimen Type	Category	PCR Panel Size								
			5	10	15	20	25	30	35	40	45
All	All	Organisms Detected	21,773	28,225	31,607	33,403	34,655	35,097	35,228	35,261	35,275
		Organisms not Detected	13,502	7050	3668	1872	620	178	47	14	0
		% Not Detected	38.3%	20.0%	10.4%	5.3%	1.8%	0.5%	0.1%	0.0%	0.0%
		*p*		<0.01	<0.01	<0.01	<0.01	0.22	0.73	0.93	0.99
	PMO	Organisms Detected	11,455	18,164	21,373	23,235	24,400	24,832	24,946	24,966	24,976
		Organisms not Detected	13,521	6812	3603	1741	576	144	30	10	0
		% Not Detected	54.1%	27.3%	14.4%	7.0%	2.3%	0.6%	0.1%	0.0%	0.0%
		*p*		<0.01	<0.01	<0.01	<0.01	0.18	0.75	0.94	0.99
	FO	Organisms Detected	3698	6165	6689	7043	7293	7435	7455	7469	7473
		Organisms not Detected	3775	1308	784	430	180	38	18	4	0
		% Not Detected	50.5%	17.5%	10.5%	5.8%	2.4%	0.5%	0.2%	0.1%	0.0%
		*p*		<0.01	<0.01	<0.01	0.08	0.46	0.89	0.95	0.99
UC+	All	Organisms Detected	14,735	18,040	20,479	21,489	22,401	22,691	22,773	22,780	22,783
		Organisms not Detected	8048	4743	2304	1294	382	92	10	3	0
		% Not Detected	35.3%	20.8%	10.1%	5.7%	1.7%	0.4%	0.0%	0.0%	0.0%
		*p*		<0.01	<0.01	<0.01	<0.01	0.07	0.86	0.98	0.99
	PMO	Organisms Detected	8401	12,175	14,496	15,654	16,468	16,777	16,861	16,867	16,869
		Organisms not Detected	8468	4694	2373	1215	401	92	8	2	0
		% Not Detected	50.2%	27.8%	14.1%	7.2%	2.4%	0.5%	0.0%	0.0%	0.0%
		*p*		<0.01	<0.01	<0.01	<0.01	0.26	0.85	0.98	0.99
	FO	Organisms Detected	1916	2994	3264	3443	3537	3605	3606	3608	3608
		Organisms not Detected	1692	614	344	165	71	3	2	0	0
		% Not Detected	46.9%	17.0%	9.5%	4.6%	2.0%	0.1%	0.1%	0.0%	0.0%
		*p*		<0.01	<0.01	0.05	0.34	0.81	0.99	0.99	0.99
UC-	All	Organisms Detected	7038	10,185	11,128	11,914	12,254	12,406	12,455	12,481	12,492
		Organisms not Detected	5454	2307	1364	578	238	86	37	11	0
		% Not Detected	43.7%	18.5%	10.9%	4.6%	1.9%	0.7%	0.3%	0.1%	0.0%
		*p*		<0.01	<0.01	<0.01	0.07	0.44	0.79	0.90	0.99
	PMO	Organisms Detected	3054	5989	6877	7581	7932	8055	8085	8099	8107
		Organisms not Detected	5053	2118	1230	526	175	52	22	8	0
		% Not Detected	62.3%	26.1%	15.2%	6.5%	2.2%	0.6%	0.3%	0.1%	0.0%
		*p*		<0.01	<0.01	<0.01	0.03	0.47	0.84	0.93	0.99
	FO	Organisms Detected	1782	3171	3425	3600	3756	3830	3849	3861	3865
		Organisms not Detected	2083	694	440	265	109	35	16	4	0
		% Not Detected	53.9%	18.0%	11.4%	6.9%	2.8%	0.9%	0.4%	0.1%	0.0%
		*p*		<0.01	<0.01	0.05	0.13	0.52	0.86	0.93	0.99

**Table 7 ijms-24-14269-t007:** Common clinical criteria for urinary PCR testing.

Recurrent UTI (2nd visit within 6 months, 3rd visit within a year)
Persistent irritative lower urinary tract symptoms (e.g., frequency, urgency, dysuria), and previous negative culture
UTI symptoms with suspicion of sexual transmitted infections
Interstitial cystitis
Prostatitis: chronic/acute

**Table 8 ijms-24-14269-t008:** Forty-five organisms and one positive control in the PCR panel.

Organisms	Fisher’s Assay IDs
*Acinetobacter baumannii*	Ba04932084_s1
* Actinobaculum schaalii *	AIPAFMX
* Aerococcus urinae *	AIQJDS5
* Alloscardovia omnicolens *	AIVI6H1
*Candida albicans*	Fn04646233_s1
*Candida glabrata*	Fn04646240_s1
*Candida parapsilosis*	Fn04646221_s1
* Chlamydia trachomatis *	Ba04646249_s1
*Citrobacter amalonaticus*	AP7DR2J
*Citrobacter freundii*	Ba04932088_s1
*Citrobacter koseri*	AIX02UH
*CMV*	Pa03453400_s1
* Corynebacterium riegelii *	AI5IRVT
* Corynebacterium urealyticum *	AI39TPL
*Enterobacter aerogenes*	Ba04932080_s1
*Enterobacter cloacae*	Ba04932087_s1
*Enterococcus faecalis*	Ba04646247_s1
*Enterococcus faecium*	Ba04932086_s1
*Escherichia coli*	Ba04646242_s1
*HSV1*	Vi04230116_s1
*HSV2*	Vi04646232_s1
*Human herpesvirus-6*	AI1RW3J
*Klebsiella oxytoca*	Ba04932079_s1
*Klebsiella pneumoniae*	Ba04932083_s1
*Morganella morganii*	Ba04932078_s1
*Mycobacterium tuberculosis*	APEPTGE
* Mycoplasma genitalium *	Ba04646251_s1
* Mycoplasma hominis *	Ba04646255_s1
* Neisseria gonorrhoeae *	Ba04646252_s1
*Pantoea agglomerans*	AP47WMK
*Proteus mirabilis*	Ba04932076_s1
*Providencia stuartii*	Ba04932077_s1
*Pseudomonas aeruginosa*	Ba04932081_s1
*Serratia marcescens*	AIMSIYA
*Staphylococcus aureus*	Ba04646259_s1
*Staphylococcus epidermidis*	Ba04230918_s1
*Staphylococcus haemolyticus*	APMFXMX
*Staphylococcus lugdunenesis*	APTZ9W7
*Staphylococcus saprophyticus*	Ba04932085_s1
*Streptococcus agalactiae*	Ba04646276_s1
* Streptococcus oralis *	AP9HJTH
*Streptococcus pasteurianus*	APCE4P6
*Streptococcus pyogenes*	AIVI6AD
*Trichomonas vaginalis*	Pr04646256_s1
* Ureaplasma urealyticum *	Ba04646254_s1
*Xeno Assay—CONTROL*	Ac00010014_a1

The fastidious organisms were highlighted in blue.

## Data Availability

The original data in this study are restricted from public access due to patient’s privacy.

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
