# Peer review of "The Essential Role of PCR and PCR Panel Size in Comparison with Urine Culture in Identification of Polymicrobial and Fastidious Organisms in Patients with Complicated Urinary Tract Infections"

_ijms, 2023, doi:10.3390/ijms241814269_

Round 1
Reviewer 1 Report
The research work entitled “The Essential Role of PCR in Identification of Polymicrobial and Fastidious Organisms in Patients with Complicated Urinary Tract Infections in Comparison with Urine Culture” by Hao et al is quite interesting and up-to-date. After going through the manuscript, I found several issues with this manuscript that must be incorporated in order to accept this manuscript for publication. My suggestions are: -
1. References are not in the format of MDPI. Modify.
2. Objective and novelty of this manuscript are not clear it must be added at the end of the introduction section.
3. All the units must be in SI format like μl =μL
Conclusion: major revision
Author Response
Dear Reviewer,
Thank you so much for your constructive comments. We have modified the manuscript in multiple aspects.
A: A more strict criteria of complicated urinary tract infection was applied for selection of the patients which lead to the exclusion of the patients not clearly following strict clinical protocols.
- We used a novel organism line-item concomitant analysis to compare the UC and PCR detection rate. This clearly helps us to understand the difference of detection rate between UC and PCR.
- We further improved the panel analysis data.
- We added PCR Panel Size to the tile to reflect our important findings in this manuscript.
We have further addressed issues raised by reviewers on a one-by-one basis in “blue” as follows.
The research work entitled “The Essential Role of PCR in Identification of Polymicrobial and Fastidious Organisms in Patients with Complicated Urinary Tract Infections in Comparison with Urine Culture” by Hao et al is quite interesting and up-to-date. After going through the manuscript, I found several issues with this manuscript that must be incorporated in order to accept this manuscript for publication. My suggestions are: -
- References are not in the format of MDPI. Modify. The reference format was modified with Chicago MIPD style.
- Objective and novelty of this manuscript are not clear it must be added at the end of the introduction section. We added the objective and novelty of this manuscript at the end of the introduction as shown below:
The current study presents a retrospective review of 40029 patients whose urine specimens were tested concomitantly by both UC and PCR with the aim to determine whether the addition of urinary PCR testing to conventional UC is of clinical utility specifically in this difficult to treat subset of the patients. In addition, we seek to determine whether and to what degree PCR panel size increase sensitivity in the detection of potential urinary tract pathogens in patients with cUTI, both overall and when PMO and FO are present.
- All the units must be in SI format like μl =μL. We reformatted μl as μL in the text.
- Conclusion: major revision. We also revised the conclusion in the text and as bellow:
The subset of patients with UTI that present with or develop cUTI represent a challenging clinical management problem, one which can result in substantial patient morbidity and consumption of significant healthcare resources. While there is ample data that PCR outperforms UC in organism detection in UTI, our study is the first to specifically analyze this differential in patients with cUTI, and to quantify this differential for patients with PMO and FO. We also found that in addition to overall positivity rates, individual organism line-item concordance strongly favored PCR over UC for virtually all organisms on the PCR panel, with nearly half the organisms on the PCR panel not isolated on any UC. Despite the superiority of PCR over UC, even a 45-organisms PCR panel did not capture 100% of organisms isolated on UC, nor was there 100% concordance for any organism found on both tests. We conclude that for patients with cUTI, UC and PCR are complementary tests that should be used in conjunction to provide clinicians with complete data on which to base diagnostic and management decisions for this difficult to treat subset of patients.
We further revised manuscript for any grammars, wording issues.
If you have any further comments, please let us know without any hesitation.
Xingpei Hao MD,PhD
Reviewer 2 Report
This is an interesting work that explores the role of PCR as a marker of polymicrobial infection. The applicability of surrogate markers of the complexity of an infectious process of the urinary tract is the most necessary in daily clinical activity. Correlating the reliability of the PCR to tests that explore the alterations of the microbiome and urinary microbiota would represent a further step forward in the management of this topic. Congratulations to the authors for the good work.
No suggestions about the language
Author Response
Dear Reviewer,
Thank you so much for your constructive comments. To increase the quality of the paper, we have modified the manuscript in multiple aspects.
A: A more strict criteria of complicated urinary tract infection was applied for selection of the patients which lead to the exclusion of the patients not clearly following strict clinical protocols.
- We used a novel organism line-item concomitant analysis to compare the UC and PCR detection rate. This clearly helps us to understand the difference of detection rate between UC and PCR.
- We further improved the panel analysis data.
- We added PCR Panel Size to the tile to reflect our important findings in this manuscript.
If you have any further comments, please let us know without any hesitation.
Xingpei Hao MD, PhD